# Volcanic evolution of an ultraslow-spreading ridge

H. H. Stubseid [1,2] ✉, A. Bjerga [1,2], H. Haflidason [1], L. E. R. Pedersen[1] & R. B. Pedersen[1]

Nearly 30% of ocean crust forms at mid-ocean ridges where the spreading rate is less than 20 mm per year. According to the seafloor spreading paradigm, oceanic crust forms along a narrow axial zone and is transported away from the rift valley. However, because quantitative age data of volcanic eruptions are lacking, constructing geological models for the evolution of ultraslow-spreading crust remains a challenge. In this contribution, we use sediment thicknesses acquired from ~4000 km of sub-bottom profiler data combined with [14]C ages from sediment cores to determine the age of the ocean floor of the oblique ultraslow-spreading Mohns Ridge to reveal a systematic pattern of young volcanism outside axial volcanic ridges. Here, we present an age map of the upper lava flows within the rift valley of a mid-ocean ridge and find that nearly half of the rift valley floor has been rejuvenated by volcanic activity during the last 25 Kyr.

Around 75% of all volcanic activity on Earth occurs at the mid-ocean ridges where tectonic plates spread apart. A central concept of seafloor spreading is that crustal accretion is confined to a narrow zone between 1 and 2 km in width[1–4]. According to this model, a young crust appears in the central part of the rift with a linear increase in crustal age moving towards the sides[5]. However, there is an increased understanding that crustal accretion can occur at a larger part of the rift valley, with a width of up to 10 km, e.g.[6]. The growth of ocean crust at the slower-spreading ridges, where the magma budget is highly variable along the ridge axis, is significantly more complicated than at faster-spreading ridges[6–9]. Ultraslow-spreading ridges are characterized by intermittent volcanism and a lack of transform faults[7] where a large part of the spreading is accommodated by tectonic activity rather than magmatism. These ridges alternate between magmatic and amagmatic segments where large offset normal faults, or detachments, exhume lower crustal and mantle rocks on the seafloor e.g.[10–12].

The 500 km long oblique spreading Mohns Ridge, in the Norwegian-Greenland Sea, between 71.2°N and 73.5°N (Fig. 1), has been extensively mapped and investigated for several decades. The ridge has a spreading rate of ~14 mm yr[-1][13] and an average sedimentation rate of 6 cm/Ka in the rift valley (Fig. 2), providing a unique age resolution to study volcanic processes on the millennial scale. The width of the inner valley floor varies between 6 and 17 km and the crustal thickness (4 ± 0.5 km) is thinner than the global average[14]. Orthogonal spreading segments are characterized by AVRs that are between 15–30 km long, and 5–10 km wide, where the volcanic activity appears to be focused[15,16]. Here, large extensional faults separate the volcanically active regions from tectonically dominated and sedimented areas (Fig. 1). Because of the oblique spreading, individual AVRs sometimes extend across the rift valley linking up with the rift-bounding faults[7,17]. Segmentation of the Mohns Ridge takes the form of lateral non-transform offsets (NTOs) between individual orthogonal spreading segments. These offsets range from 10 to 40 km and separate areas of the ridge that experience different volcanic and tectonic modes of spreading.

Direct constraints on the age and evolution of the crust at mid-ocean spreading ridges from the dating of single rock samples using U-series disequilibrium and zircon U-Pb dating[18–26], [40]Ar/[39]Ar dating of MORB e.g.[27] and [14]C dating of basal sediments[28,29], have significantly increased our understanding of magmatic and tectonic processes occurring during crustal accretion. Yet, there is a lack of age data, with

[1]Center for Deep Sea Research and Department of Earth Science, University of Bergen, Allégaten 41, N-5007 Bergen, Norway. [2]These authors contributed equally: H. H. Stubseid, A. Bjerga. ✉e-mail: havard.stubseid@uib.no

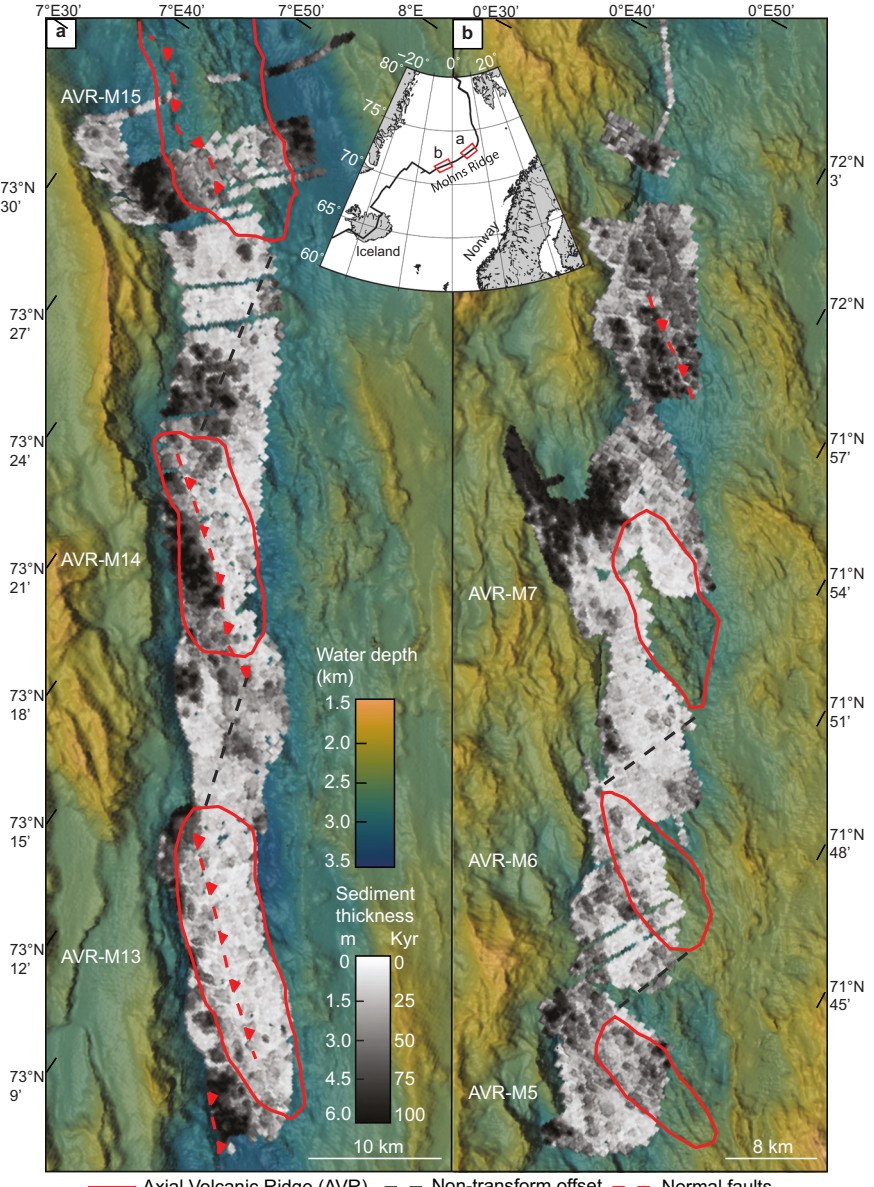

**Fig. 1 | Distribution of sediments across the rift valley floor presented as an isopach map.** The calculated sediment thickness from sub-bottom profiler (SBP) data (grey scale) is gridded at 175 m resolution and draped onto 70 m resolution bathymetry of the rift valley. The greyscale indicates the thickness of sediments and corresponding age of the underlaying lava flows based on the acquired sedimentation rate (Fig. 2 and Supplementary Information). Outline of individual axial volcanic ridges (AVRs) are marked as a solid red line. Red dashed lines indicate normal faults and black dashed lines are the non-transform offsets in between orthogonal spreading segments. **a** represents the northernmost part of the surveyed area, whereas **b**. is the southern segment of the ridge.

a resolution better than that of magnetic anomalies, from the rift valley of mid-ocean ridges that could provide a more complete view of how ocean crust accretes.

In this contribution, we present an approach to enhance our understanding of the spatial-temporal distribution of volcanism at a mid-ocean ridge. To advance our understanding of the processes and timescales involved in the formation of the ocean crust, we present a segment-scale map showing the age distribution of the upper lava flows throughout the rift valley floor. Sediment thicknesses, calculated from more than 4000 km of sub-bottom profiler data, are converted to age by establishing accurate sedimentation rates from the [14]C dating of handpicked foraminifera in sediment cores (details in Supplementary Information). In this work, we reveal that the volcanic activity is not restricted to pronounced axial volcanic ridges (AVRs) but occurs throughout the width of the rift valley floor.

## Results
### Constraining the timing and distribution of volcanism
To investigate the volcanic history of the rift valley, we report the results from ~4000 km of seismic lines that were collected by autonomous underwater vehicles (AUV) carrying sub-bottom profilers. The dataset covers an area of 1500 km² that represents approximately 25% of the inner rift valley floor of the Mohns Ridge (Fig. 1). We tracked continuous seismic lines in transects perpendicular to the rift valley with a spacing of 200 to 500 m between individual lines (Supplementary Fig. S1). In areas with no or very thin sediment cover (less than 30 cm) we used high-resolution (1 m) bathymetry and backscatter data to constrain the sediment cover. Based on these data, the variable thickness of hemipelagic sediment that is present on top of the volcanic basement is reported as an isopach map in Fig. 1 (see Supplementary Information for further details).

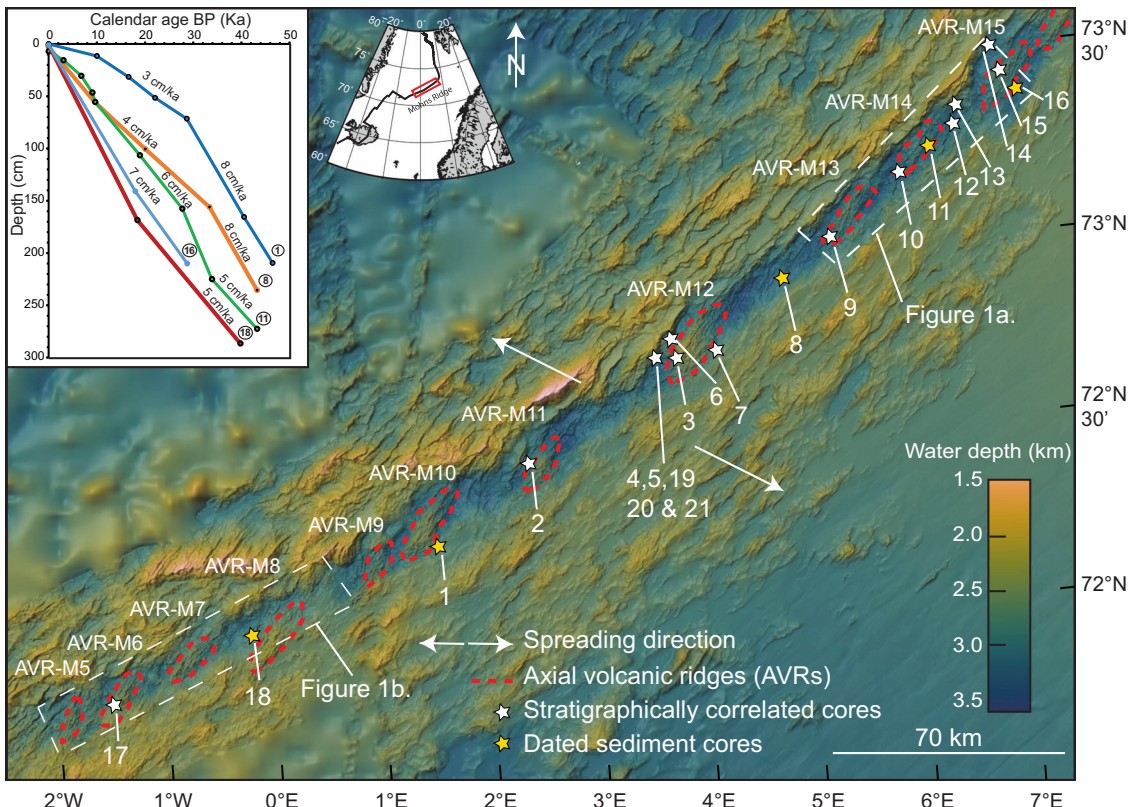

**Fig. 2 | Overview of collected sediment cores and calculated sedimentation rates.** The collected sediment cores are marked as stars on a 70 m resolution bathymetry with 2 times vertical exaggeration. Yellow stars are cores that have been dated and white stars are cores that are photo-scanned and analysed using X-ray fluorescence element scanning. Their chemo stratigraphy has been correlated with the dated cores to strengthen the age model (details in Supplementary Information). The outline of the axial volcanic ridges (AVRs) is marked with a red dashed line and numbered according to their location along the Mohns Ridge. Inset in the upper left corner shows the sedimentation rate for each of the dated cores, with a number corresponding to the location on the map (see Table S1 in the Supplementary Information). The average sedimentation rate is estimated to be 6 cm/Kyr for the entire rift valley.

To constrain the time dimension, sedimentation rates have been calculated along the entire length of the study area (Fig. 2 and supplementary) by [14]C dating of forams from 5 sediment cores and chemo stratigraphic correlation with the remaining 16 cores. The average sedimentation rate in the rift valley (6 cm/Kyr) is twice as high as in the open ocean of the Norwegian-Greenland Sea[30–33]. The high sedimentation rate allows for a high-resolution age correlation across the rift valley. The seismic acquisition parameters had a true vertical resolution of 20–30 cm (see details in Supplementary Information), yielding an age resolution of close to five thousand years at the time-averaged sedimentation rates. After sediments are deposited, they can be reworked through mass wasting or by bottom current activity. To avoid bias due to the effects of gravitational mass wasting we exclude all sediment thickness measurements associated with slopes above 30 degrees. There are no signs of mass wasting/slumping within our chirp profiles (Fig. S2) or gravity cores (Figs. S9 & S10) suggesting that the sediment cover is a result of steady hemipelagic sedimentation. To our knowledge, there are no measurements of bottom currents along the Mohns Ridge. Only intermediate currents in the water column have been studied e.g.[34] but their effect on the seafloor remains poorly understood. Also, modern-day current patterns may not be representative from a geological time perspective. The chirp profiles document that sediments are evenly draped on top of the seafloor topography (Fig. S2) with no systematic differences in the degree of sedimentation between local basins and bathymetric highs indicating weak bottom currents. As we show that the hemipelagic sediment accumulates at a relatively constant rate, at least for the last 50 Kyr, (Fig. 2 and Supplementary Information), the sediment thickness reflects the age of the underlying volcanic flows[35]. The isopach map (Fig. 1), accordingly, reflects the spatial-temporal evolution of volcanic activity within the rift valley.

## Spatial-temporal volcanic evolution of the rift valley

The thickness of sediments above the lava flows is generally less than 9 meters and only occasionally reaches 11 meters. These thickness measurements imply that within the rift valley, the upper lava flows of the volcanic basement are typically younger than 180 Kyr. The analysis of data and isopach map further reveals that approximately 50% of the surveyed region is covered with sediments less than 1.5 meters thick, corresponding to an age of 25 Kyr (Fig. 3).

The distribution of sediment thicknesses (mean = 2.7 m and median = 1.6 m) demonstrate that the rift valley floor is dominated by thin sediments and young volcanic crust. This suggests that within most of the rift valley, the older lava flows and sedimentary deposits are being continuously covered by younger eruptions.

We define a total of 16 AVRs as pronounced bathymetric highs along the Mohns Ridge. These volcanic structures comprise 30% of the rift valley floor and have an average surface area of 125 km². A total of 7 AVRs have been surveyed as part of this study, and 35% of our data points are within these sub-areas (Fig. 3a). Only one AVR (AVR-M5) is interpreted to be volcanically extinct as the isopach map shows an average sediment thickness of 2–3 m, corresponding to an average age of 30–50 Kyr, for the entire surface of the structure (Fig. 1). All other surveyed AVRs exhibit a surface dominated by sediment thicknesses below 1.5 and are classified to be volcanically active.

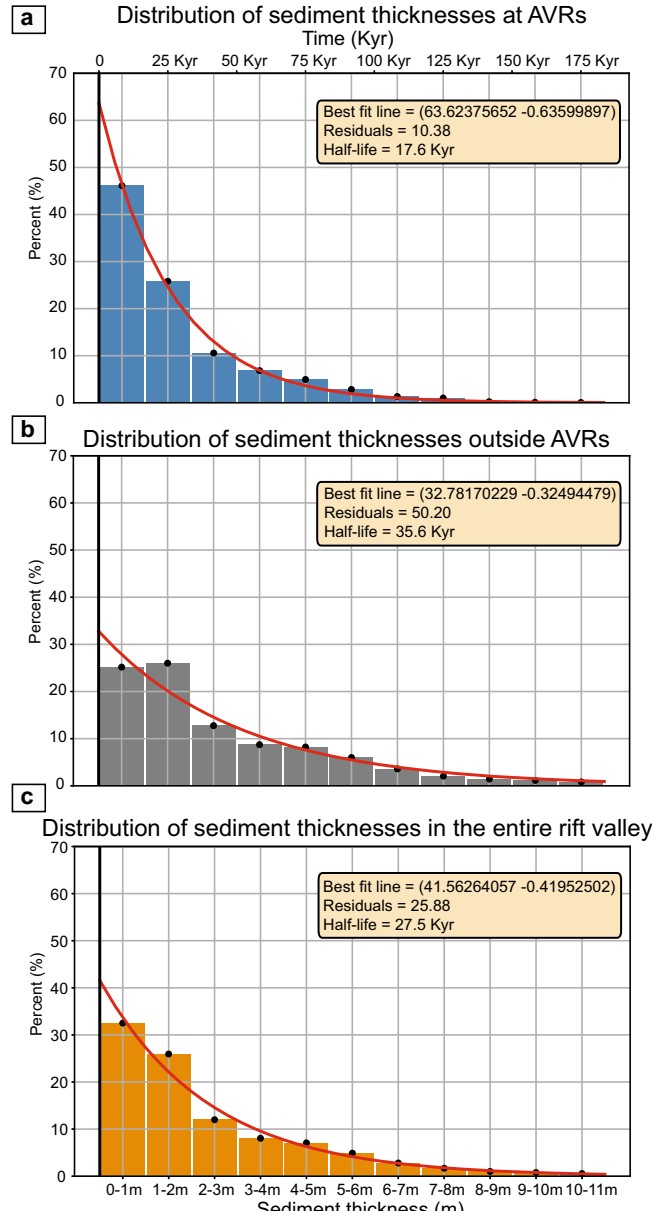

**Fig. 3 | Best-fit regression line through the sediment distribution.** The figure shows the distribution of the sediment thickness measured from the sub-bottom profiler (SBP) data with a bin size of 1 meter. **a**. is the distribution within axial volcanic ridges (AVRs), **b**. is outside AVRs, and **c**. is these two combined for the entire rift valley. All thickness measurements above 11 meters have been removed as these measurements are from areas outside the rift valley. Areas within the survey lines that lack thickness measurements are given dummy values (0 m) and all data points in areas with slopes above 30 degrees are removed (details in Supplementary Information). The data sets are represented by exponential regression lines in red, and accompanying the lines are inset boxes containing the derived constants, residuals, and half-life values.

Individual volcanic centers can be seen in the isopach map as small, isolated fields with varying sediment cover, leaving a patchy age pattern (Fig. 4a and b). The volcanic structures forming this patchy pattern have an average size of ~1–2 km² and represent the unit cell of the volcanic sequence (Fig. 4b). The isopach map also shows that the volcanic activity is partly controlled by major normal faults that nucleate at the AVRs, for later to develop into the major fault systems that define the rift valley. The faults appear to inhibit or cut off the volcanic activity on the footwall side of the faults (Fig. 1). For example,

at AVR M-14 (Fig. 4d), the isopach map reveals a volcanically active zone, 4 to 5 km wide on the central part of the AVR, limited to the west by an active normal fault and a volcanically inactive area covered in 4–6 m of sediments.

Calculations show that 62% of the total AVR surfaces are younger than 25 Kyr (Fig. 3a). The total surveyed AVR surfaces have a mean sediment thickness of 1.6 m (= 27 Kyr) and a median of 1.1 m (= 18 Kyr). The abundance of thicker sediments is rapidly decreasing, and sediment thicknesses above 3 m are nearly absent at the active AVRs (Fig. 3a). The sediment distribution follows here an exponential pattern. The exponential best-fit curve shows a near-perfect match with the data (10% residuals) and reveal a statistical half-life of 18 Kyr for the AVR surfaces. This implies that within 18 Kyr, half of the AVR surfaces have been covered by younger volcanic flows. It also suggests that the volcanic renewal of the AVRs has taken place at about the same rate for the last 150 Kyr.

The oldest volcanic seafloor is found in the deep parts of the rift valley that flank the AVRs, or within NTOs. Here, the volcanic seafloor is locally covered by up to 11 m of sediments, corresponding to an age of 180 Kyr. Calculations show that thicker sediments (from 3 m to 11 m) are more abundant than on AVRs (Fig. 3b). The isopach map also reveals a patchy volcanic pattern like that seen on the AVRs (Fig. 4c). The areas flanking the AVRs and the NTOs show a similar age pattern (Fig. 3b), different from the AVRs. The sediment thicknesses reveal a large age span with an average age of 55 Kyr for the upper lava flows outside AVRs. The sediment thickness distribution outside AVRs reveals a mean value of 3.3 m (= 55 Kyr) and a median of 1.9 m (= 32 Kyr) clearly illustrating the abundance of thicker sediments and older upper lava flows. The best-fit regression line (Fig. 3b) gives a half-life of ~36 Kyr, twice as high as for the AVRs. This suggests that areas outside AVRs can maintain older lava flows at the surface and that the volcanic surface renewal is significantly slower than for the AVRs. However, the best-fit line has 50% residuals indicating a weak match between the line and the actual data distribution. This means that the volcanic age pattern is not solely a result of volcanic renewal at a constant rate.

Young volcanic constructs are common also in these areas and data analysis reveals that 40% of the areas outside AVRs are younger than 25 Kyr (Fig. 3b). These young volcanic events occur from the foot of the AVRs to the bounding faults of the inner rift valley - where young lava flows have ponded against the major faults, extruded on top of the footwall (Fig. 5), and locally been uplifted as rider blocks. The isopach map and sediment thickness statistics reveal that for the volcanic seafloor that formed during the last 25 Kyr, more than 50% formed outside the AVRs.

## Discussion

The distribution of sediments together with high-resolution bathymetry and backscatter data along the Mohns Ridge provides quantitative age constraints for the volcanic seafloor and shows that volcanism occurs across the width of the rift valley floor (Fig. 1 and Fig. 5). The observed width of the volcanically active zone, here defined as the zone of volcanism younger than 25 Kyr, varies from 2 to >10 km along the ridge. This is wider than the 1–2 km wide spreading axis observed at a segment of the Mid-Atlantic Ridge[2]. There, the spreading axis is defined as the area of active erupting fissures (primary eruptive vents) fed by underlying dikes, and eruptions occurring outside the spreading axis are interpreted to represent secondary vents with lava transport through tubes and channels[2].

Along the AVRs at the Mohns Ridge, we see no sign of systematic age patterns that could support volcanic accretion along a relatively narrow zone and that the volcanic seafloor gets gradually older away from such a spreading axis. Close to the AVRs, we cannot exclude that eruption may be secondary and fed by lava transported down-slope through tubes and channels. But, far away from the AVRs, at NTOs, and

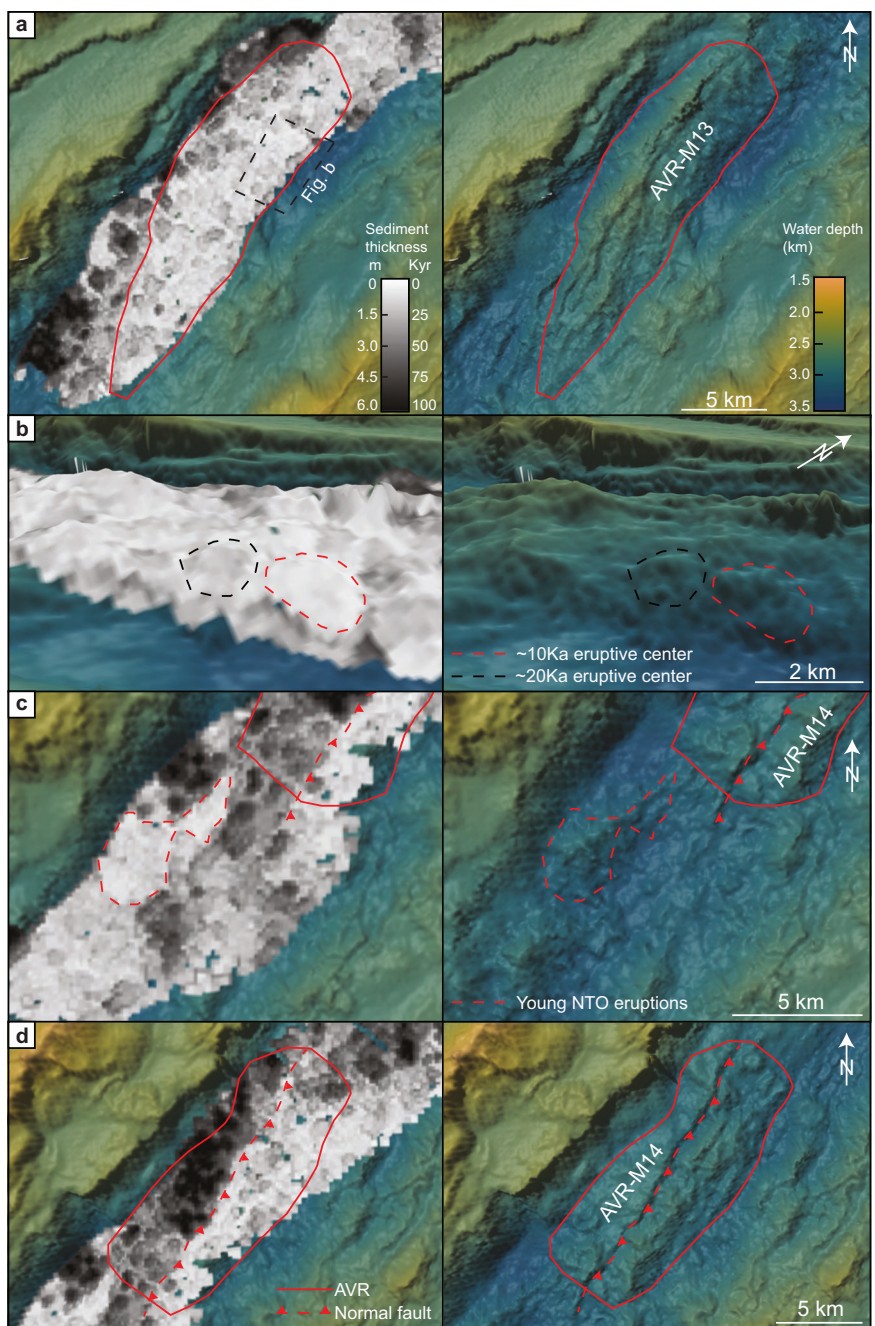

**Fig. 4 | Detailed overview of the sediment distribution in the rift valley of the Mohns Ridge. a** Sediment distribution at AVR-M13 reveal a patchy pattern interpreted to represent the unit cell of the volcanic sequence, alongside 70-m resolution bathymetry for the same area. The sediment thickness is indicated by the greyscale and is converted to age based on the sedimentation rate of 6 cm/Ka (Fig. 2). The outline of the axial volcanic ridges (AVRs) is marked as a solid red line. **b** Detailed oblique view of AVR-M13 showing both the sediments thickness and clean bathymetry on the side. Here, subtle variations in the sediment thickness can be seen. Two volcanic centers, with an approximate age difference of 10 Kyr, is highlighted to further document the unit cells of the volcanic sequence. **c** Detailed view of the non-transform offset between AVR-M13 and M14 show a similar patchy age pattern with large internal variations. Also here, both the sediment thickness and clean bathymetry is presented. A young volcanic construction within the non-transform offset (NTO) is highlighted with a red dashed line. **d** Sediment distribution and clean bathymetry for AVR-M14. Here, a normal fault partly controlling the volcanism can be seen in both datasets. On the eastern side of the fault, we observe only thin sediments and a young volcanic crust, whereas the eastern side is more heavily sedimented.

in deep areas next to the rift valley inner bounding faults (Fig. 5), a source from local magma feeding systems seems more likely[8,9,36].

Given a spreading rate of 14 mm/yr[13] and a width of the inner rift valley floor of ~10-15 km, the age of the crust near the rift valley inner bounding faults should be around 1 Myr on average. In contrast, our isopach map (Fig. 1) reveals that the age of the uppermost lava flow along the rift valley floor of the Mohns Ridge rarely exceeds 180 Kyr. Within the bounds of our dataset, the entire surface has been renewed

by volcanic activity since the Marine Isotopic Stage 6 (MIS 6). This means that the top lava flow within the rift valley is much younger than the crustal age predicted based on the linear spreading of the plates at constant rates and a narrow axial-centric spreading axis[2,24]. By extrapolating our dataset to the entire rift valley of the Mohns Ridge, it becomes clear that approximately half the inner rift valley floor has been renewed by volcanism during the last ~25 Kyr. Therefore, factors other than spreading rate may be important in determining the

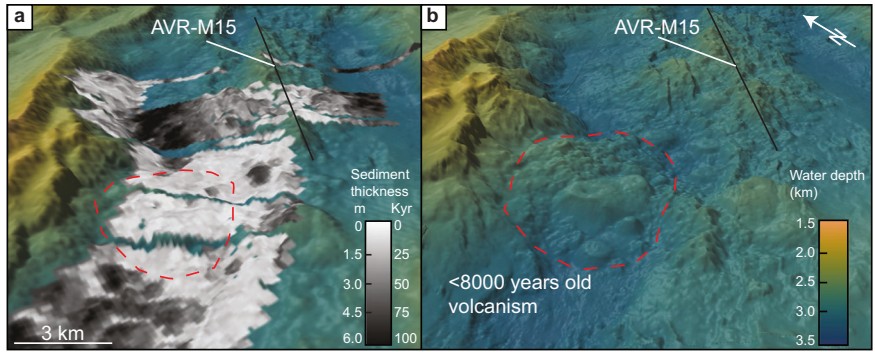

**Fig. 5 | Example of recent volcanism at the rift bounding fault. a** Sediment distribution and age of the volcanic crust for the northernmost part of the surveyed area. The sediments demonstrate a young terrain up against the fault zone, with older and more sedimented seafloor both to the north and south. The black line indicates the size and orientation of axial volcanic ridge (AVR). The figure is shown in an oblique view. **b** High-resolution ship bathymetry (15 m resolution), with 2 times vertical exaggeration, for the same area as in Fig. 5a. The bathymetry highlights the volcanic terrain and demonstrates the construction of circular volcanoes (1–2 km in diameter) and a hummocky terrain produced from a primary eruptive vent extruding at the footwall of the rift bounding fault.

production and renewal of oceanic crust at the slowest-spreading ridges[37,38].

Our findings document that the volcanism follows different patterns at AVRs than for areas outside. Volcanism at AVRs is surprisingly robust through time with a high renewal rate, twice as high as outside. Our documentation of young volcanism at nearly all studied AVRs, and a systematic pattern of even volcanism through time, suggest that the AVRs have been a steady-state volcanic system at least for the last 150 Kyr. This indicates a constant heat flow and melt supply towards all AVRs making them highly volcanically robust and productive. The fact that they stay volcanically active at the same time suggests that the volcanic activity is not jumping from place to place along the rift valley but stays constant at the same place for a long time. We see no indication of waning and waxing stages of volcanism[37] at these ridges within our time interval and argue that the volcanism has occurred at a constant rate. It has been proposed that external forcings such as changes in sea level, glaciations/deglaciations, or orbital variations[39–43] could lead to changes in the rate and amount of volcanism at mid-ocean ridges. Although there have been significant glaciations/deglaciations and sea level fluctuations in the Arctic region within the past 150 Kyr, our data do not demonstrate any variation that could be attributed to such external forces. This suggests that the magmatic system of the AVRs has remained stable and impervious to these external influences.

Volcanic activity outside AVRs follows a slightly different and more infrequent pattern. Here, we see larger age variations in the upper lava flows, indicating more pulsating volcanism through time. This could be explained by periods of more intense volcanism, and potentially larger eruptions, alternating with periods lacking volcanism in these areas. It is also possible that the magmatic system in these areas is less robust and more sensitive to external or tectonic forcing. These areas occur next to the major fault-system bounding the axial valley, or within the non-transform offset zones affected by lateral shear. The deviation from the regular exponential pattern seen at the AVRs may therefore reflect a stronger tectonic influence on the volcanic activity. Even though the volcanic productivity is lower outside AVRs (Fig. 3b), the total areal emplacement of lavas in these areas exceeds that of AVRs. Therefore, areas outside AVRs contribute significantly to the total production of oceanic crust and account for a substantial amount of the overall magma budget along the ridge.

Volcanic activity is the main driver for hydrothermal activity, and active hydrothermal vent fields have been discovered from within the rift valley of all slow and ultraslow-spreading ridges[44–49]. These vent fields support diverse and unique faunas and are associated with potentially economically valuable mineral deposits. Slow and ultraslow-spreading ridges are generally suggested to sustain long-lived hydrothermal systems and produce the largest mineral deposits[44,50–52]. However, the rapid and widespread volcanic surface renewal documented in this study place constraints on the preservation of mineral deposits on the seabed. Our estimates of renewal rates within the rift valley suggest that hydrothermal vents outside AVRs are likely to live longer and produce larger mineral deposits preserved at the seafloor. However, with a constant heat flow towards AVRs, the majority of hydrothermal vent fields are expected to be located here. Based on the half-life ($t_{1/2}$) of 18 Kyr for these areas, there is a 50% chance that any mineral deposits that formed before this time are buried after one $t_{1/2}$. Furthermore, because the amount of older seafloor surfaces decreases exponentially, it is highly unlikely that mineral deposits older than 100 Kyr are exposed. From an economic geology perspective, we, therefore, suggest that mineral deposits that form at the inner rift valley floor are likely to be buried underneath lavas making them difficult to detect and even harder to exploit. Therefore, exploitation for mineral deposits that do not consider the rapid and widespread surface renewal will likely overestimate the availability of sulfide deposits in the rift valley.

Volcanic rejuvenation of the oceanic seafloor at mid-ocean ridges is both a constructive and destructive event that has important implications for many aspects of ridge accretion. The determination of volcanic seafloor ages from sediment thickness, with a resolution of ~8Ka, along the Mohns Ridge greatly expands our understanding of spreading dynamics and volcanic processes along the slowest spreading ridges of the global ridge system. We argue that the pattern of volcanism and crustal accretion presented here is representative of larger parts of the global ridge system with several interesting consequences for the understanding of ultraslow spreading-ridge environments. First, crustal accretion occurs on a longer timescale than previously believed[1,2,20,21]. Hence, axial-centric spreading models do not accurately describe crustal accretion at ultraslow spreading ridges[24]. Second, primary eruptive vents occur within the entire width of the rift valley floor suggesting that the spreading axis is not only defined by the AVRs but that spreading events and volcanism can occur anywhere within the rift valley. Third, volcanism is more abundant and frequent at AVRs than outside suggesting a focused melt delivery toward these areas. However, as AVRs comprise only 30% of the rift valley floor the total emplacement of lava flows follows a near 1:1 ratio between AVRs and areas outside. Fourth, young volcanism is widely distributed through numerous small eruptive areas. Consequently, the number of volcanic eruptions may be underestimated at slow- and ultraslow-spreading ridges[53]. Finally, our contribution provides a volcanic age map of the rift valley floor of an entire ridge segment and reveals that volcanism is widely distributed in time and space at ultraslow-spreading ridges.

## Data availability

The processed acoustic data is available at the Norwegian Petroleum Directorate: https://kartkatalog.geonorge.no/metadata/dyphavsdata/fe943f30-9a69-4c9c-9f35-726e28d9126f Due to extensive amounts of data, acoustic data is not currently available for direct download. However, the data can be ordered from the Norwegian Petroleum Directorate at a reasonable request as described here: https://www.npd.no/globalassets/1-npd/om-oss/informasjonstjenester/karttjenester/bestilling-av-dyphavsdata_en.pdf The $^{14}$C ages generated in this study are provided in the Supplementary Information.

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

## Acknowledgements

The crew on RV G.O. Sars and the Ægir6000 ROV team are acknowledged for their assistance during a series of cruises. We thank Jarle A. Vikebø, Alden Denny, Stig Monsen, Rasmus Rikter-Svendsen, Markus Mila, Lubna S. J. Al-Saadi, and Heidrun M. Sande for contributions to data and sample processing, and Thibaut Barreyre and Ryan Portner for valuable feedback on previous drafts of the manuscript. The Norwegian Petroleum Directorate shared acoustic data for the study. Funding was provided through a grant to Centre for Deep Sea Research from the K.G. Jebsen and the Trond Mohn Foundations.

## Author contributions

R.B.P. and H.H.S. designed the study with input from A.B. and H.H. R.B.P. secured data and funding for the project. H.H.S. and A.B. interpreted acoustic data. H.H. lead the processing of sediment cores and handled 14C dating and L.E.R.P. performed data analysis. H.H.S. and A.B. wrote the manuscript with input from H.H., L.E.R.P., and R.B.P.

## Competing interests

The authors declare no competing interests.
