## [Peer Review File · Nature Communications]

Volcanic evolution of an ultraslow-spreading ridgeREVIEWER COMMENTS

Reviewer #1 (Remarks to the Author):

General comments

In this manuscript, Stubseid et al. present a near-bottom geophysical and geological study of an ultraslow-spreading segment of the Mohns Ridge. They use a combination of sub-bottom (chirp) profile images collected using an autonomous underwater vehicle combined with bathymetric data and dated sediment gravity cores to attempt to estimate the timing of volcanism with the rift valley zone. The chirp images are used to estimate sediment thickness, while age and thus sedimentation rate estimates are obtained from ^{14}C dating of foraminifera in the cores. Combined together, the authors then attempt to map recent volcanism across the rift valley floor, and find that recent volcanism (<25 kyr in their definition) is distributed across a broad zone ranging from 2 to >10 km away from the axis. Based upon the distribution of regions of thin or absent sediment, Stubseid et al. infer that volcanism could occur anywhere in the rift valley, and not just at axial volcanic ridges, and then speculate upon the implications for the distribution of hydrothermal vents and mineral deposits on ultraslow-spreading ridges. While the author's approach has promise, in its current form the conclusions presented in the Ms are not well supported, due to three main problems outlined below. I have also provided some more minor suggestions that I hope the authors will find helpful.

The first major problem is with lack of details given about the data acquisition, processing, and interpretation. Unfortunately, the chirp images, which are the most important dataset that underpin the study, are not properly presented (only small snippets are shown in Supplementary Figures S2-S4, although without appropriate length or depth scales and therefore cannot be evaluated properly).

Tracklines showing where the chirp images were collected are not plotted, and the processing description given in the Supplement does not give necessary basic details. For example the model of the instrument, the acquisition frequency sweep, details of any filters applied, or whether the authors used the envelope or analytic signal, are all missing. Details are also lacking on what velocity was used to convert from two-way travel time to depth, and the characterization of uncertainty in sediment thickness estimates is poorly defined (on page 2 of supplement). Several issues remain unclear, for example what is the acoustic character of sedimentary section? How variable is the basement pick – is it more diffuse where the basement is highly uneven, for example? Other data (multibeam bathymetry, backscatter) are partially presented in 3d views in the figures which are difficult to evaluate, while remotely operated vehicle (ROV) imagery, which the authors state provides ground-truthing, is not included in the figures.

Secondly, the chirp data tracklines were acquired at a spacing of 200-800 m (note it is not possible to evaluate the spacing since the tracklines are not plotted on the figures). In order to generate sediment isopach maps, the 2d sediment estimates were gridded at a resolution of 175 m (caption for Figure 1; Line 70). Since the grid spacing is potentially much less than the data spacing, the interpolation scheme (which is not specified, and needs more detail in any case) is likely to generate significant artifacts. The authors then use the interpolated sediment thickness grids as images to generate thickness statistics in Figure 3, which is likely to be problematic since the choice of interpolation scheme and grid resolution will impact the results. It would be better to analyze the population of individual sediment thickness estimates from the chirp profiles directly, without the interpolation.

Thirdly, post-depositional processes, such as mass-wasting, current-driven erosion or redeposition, that likely significantly impact their sediment thickness, are not discussed. These processes have the potential to change the locus of sediment on the seafloor, especially in places like the Mohns ridge axial valley, where local basins and highs can lead to major differences in sediment accumulations. The authors need to address this uncertainty, possibly by examining local deep-water current patterns, and the potential effects that local depositional setting and seafloor slope could have in biasing their results (e.g. axial volcanic ridge flanks vs. local basins).

Minor comments

L24 This width is thought to be more like 10 km by some (e.g. Katz et al., 2006), meaning that it's not very surprising to have volcanism anywhere in the axial trough.

L25-28 This description is an over-simplification of what most think is happening at the axis on slow-spreading ridges and could benefit from some more references and details (e.g. Bickert et al., 2020; Reston, 2018)

L32 Missing reference to other detailed geochemical work, for example using using Po isotopes (Rubin et al., 1994)

L61, L109 what is "high resolution"?

L67 Needs additional caveats – assuming that no post-depositional reworking or mass wasting occurred

L74 why compare to sedimentation rates in the equatorial MAR? Are there any other cores available on Mohns Ridge?

L77 How did you calculate the 30 cm vertical resolution of the chirp data? I would expect it to be much better than this since the instrument was on an AUV, bt impossible to check without better details in the supplement.

L83 define XRF

L105, L158 Here ROV ground truthing and visual observations are mentioned, but a detailed description of the images is missing, and images are not included

L121-126 I can't easily verify that "that the volcanic activity is partly controlled by major normal faults" from the figures as presented

L185-189 the dating of basaltic dike intrusions is interesting, but would be better to be backed up with more than just an 'in prep' reference

L190-L198 This section about hydrothermal vents and mineral deposits is speculative and I'm not sure whether it's even relevant

L201-203 This sentence is a bit of a stretch

The Ms has a series of typos throughout, see examples on L85, L197 and L208 – there are likely more.

Availability of data and materials: I could not access the data via the link provided

Figure 1

Location of chirp lines needs to be shown

Need latitude and longitude marks on this and all other maps

How are the AVR's picked? Are they really as linear/straight as shown?

How do the authors know that the faults (marked in red) are active? Are they that straight?

Inset location map is too small to be useful

Figure 3

Is this an analysis of the interpolated sediment thickness, or of the sediment thickness estimates themselves (i.e. without interpolation)? It should be the latter, since the interpolation is potentially misleading. What are the uncertainties?

Figures 4 & 5

I don't find these 3d perspective views very helpful in following the author's narrative

References Cited

Bickert, M., Lavier, L., & Cannat, M. (2020). How do detachment faults form at ultraslow mid-ocean ridges in a thick axial lithosphere? *Earth and Planetary Science Letters*, 533, 116048. <https://doi.org/10.1016/j.epsl.2019.116048>

Katz, R. F., Spiegelman, M., & Holtzman, B. (2006). The dynamics of melt and shear localization in partially molten aggregates. *Nature*, 442(7103), 676–679. <https://doi.org/10.1038/nature05039>

Reston, T. J. (2018). Flipping detachments: The kinematics of ultraslow spreading ridges. *Earth and Planetary Science Letters*, 503, 144–157. <https://doi.org/10.1016/j.epsl.2018.09.032>

Rubin, K. H., Macdougall, J. D., & Perfit, M. R. (1994). ²¹⁰Po-²¹⁰Pb dating of recent volcanic eruptions on the sea floor. *Nature*, 368(April), 841–844.

Reviewer #2 (Remarks to the Author):

The submitted manuscript “Volcanic evolution of an ultraslow spreading ridge” presents new data constraining the bathymetry, sediment thickness, and sedimentation rates along the Mohns Ridge. The authors use the data to demonstrate that the width of axial volcanism is greater than expected in this ultraslow setting. I found the manuscript to be well written and interesting, and consider the study an important contribution towards understanding the dynamics of crustal accretion at slow spreading ridges. I recommend its publication, with only minor revisions.

My most significant comment is that I found it hard to independently assess some of the data interpretations, based on the current figures. In particular, I wonder if there is a way to make Figure 1 clearer, although I recognize the challenges of presenting different data types over a large area. I make more detailed suggestion on this point and others below.

Detailed comments:

Lines 16–18: “We present the first age map of a mid-ocean ridge and find that nearly half of the 6-17 km wide inner rift valley floor has been rejuvenated by volcanic activity during the last 25 Kyr.”

I understand what the authors are trying to say here, but I do not think it is accurate to say this is the first age map of a mid-ocean ridge. Magnetic anomaly maps could be considered age maps, and there have also been other published studies with multiple dated samples along a ridge segment. For example:

Connor et al., *Nature Communications*, 2021, Thermochemical anomalies in the upper mantle control Gakkal Ridge accretion.

Baines et al., *EPSL*, 2008, The rate of oceanic detachment faulting at Atlantis Bank, SW Indian Ridge.

Schwartz et al., *Science*, 2005, Dating the growth of oceanic crust at a slow-spreading ridge.

Line 23: There is a typo, replace “were” with “where”.

Line 42: It would be helpful to describe the location of the ridge, for non-specialist readers.

Lines 65–68: Given that this text refers to the age data, the authors could consider moving it below the discussion of their age data (lines 71–78).

Lines 106–108: “Individual volcanic centers can be seen in the isopach map as small, isolated fields with varying sediment cover leaving a patchy age pattern (Fig. 4).”

These are not apparent to me in Figure 4. From the subsequent text, it sounds like this observation is based on a detailed study that the authors are currently writing up, but this does not seem to me to be supported by the data in this manuscript.

Line 122: There is a typo, replace "nucleates" with "nucleate".

Line 142: There is a typo, replace "occurs" with "occur".

Lines 145–146: "The isopach map reveals that for the volcanic seafloor that formed during the last 25 Kyr, as much as 50% formed outside the AVRs."

This is hard to assess based on the current figures (please see my comments below).

Figure 1: The manuscript figures are very high quality, but I wonder if there is a clearer way to present the combined bathymetric and sediment thickness data. The AVRs show up very nicely on the bathymetry, as seen in Figure 2. It would be useful to be able to clearly identify those and other bathymetric features in Figure 1, but it is difficult because the sediment thickness data masks the bathymetry. I suspect the authors have already experimented with this, but would it be clearer to show the bathymetry as a grayscale hillshade, then overlay that with a semitransparent color scale for sediment thickness? This might also more clearly accentuate variations in the sediment thickness, since it is difficult to see subtle differences on the current grayscale. It would also be helpful to number the AVRs in this figure, so the reader can more easily relate the descriptions in the text to the figure. Finally, I would recommend adding a label for Norway and Iceland to the location map, to provide context.

Figure 2. I would change the "processed sediment cores" label in the key to "undated cores". My initial interpretation (before reading the figure caption) was that these were cores that the authors processed for geochron., but which did not yield data, which is incorrect.

Overall, this is an interesting study and a well written manuscript, with high quality figures.

I hope these comments are useful.

Reviewer #1 (Remarks to the Author):

General comments

In this manuscript, Stubseid et al. present a near-bottom geophysical and geological study of an ultraslow-spreading segment of the Mohns Ridge. They use a combination of sub-bottom (chirp) profile images collected using an autonomous underwater vehicle combined with bathymetric data and dated sediment gravity cores to attempt to estimate the timing of volcanism with the rift valley zone. The chirp images are used to estimate sediment thickness, while age and thus sedimentation rate estimates are obtained from ^{14}C dating of foraminifera in the cores. Combined together, the authors then attempt to map recent volcanism across the rift valley floor, and find that recent volcanism (<25 kyr in their definition) is distributed across a broad zone ranging from 2 to >10 km away from the axis. Based upon the distribution of regions of thin or absent sediment, Stubseid et al. infer that volcanism could occur anywhere in the rift valley, and not just at axial volcanic ridges, and then speculate upon the implications for the distribution of hydrothermal vents and mineral deposits on ultraslow-spreading ridges. While the author's approach has promise, in its current form the conclusions presented in the Ms are not well supported, due to three main problems outlined below. I have also provided some more minor suggestions that I hope the authors will find helpful.

Response: We thank the reviewer for the detailed comments and have substantially edited our manuscript to resolve all the reviewers' concerns.

The first major problem is with lack of details given about the data acquisition, processing, and interpretation. Unfortunately, the chirp images, which are the most important dataset that underpin the study, are not properly presented (only small snippets are shown in Supplementary Figures S2-S4, although without appropriate length or depth scales and therefore cannot be evaluated properly).

Response: All relevant details about data acquisition, processing, and interpretation is now included in the supplementary information. Thanks for letting us know that the snippets of the chirp images are not sufficient. We have added a new figure in supplementary (Fig. S2) showing 5 approximately 1 km long seismic lines from within the rift valley. The lines have been chosen to represent various locations within the rift valley with varying degrees of sedimentation. The figure has an appropriate length and depth scale, and we hope that this is sufficient for the reviewers, and other readers, to evaluate our interpretations independently.

Tracklines showing where the chirp images were collected are not plotted, and the processing description given in the Supplement does not give necessary basic details. For example the model of the instrument, the acquisition frequency sweep, details of any filters applied, or whether the authors used the envelope or analytic signal, are all missing.

Response: Thank you for highlighting these needs to specify the data collection and processing. This has been addressed and improved in the supplementary. Track lines are shown in Supplementary Fig. S1. The model of the instrument used was EdgeTech

2205 with an acquisition frequency sweep of 1-9 kHz. No filters were applied, and we used the envelope signal.

Details are also lacking on what velocity was used to convert from two-way travel time to depth, and the characterization of uncertainty in sediment thickness estimates is poorly defined (on page 2 of supplement). Several issues remain unclear, for example what is the acoustic character of sedimentary section? How variable is the basement pick – is it more diffuse where the basement is highly uneven, for example?

Response: We have specified that we used a velocity of 1500 m/s to convert from two-way travel time to depth. Also, we have elaborated on the acoustic character of the sedimentary section, the basement pick, and therefore the uncertainty of the calculated sediment thicknesses. This is further described in the supplementary material.

Other data (multibeam bathymetry, backscatter) are partially presented in 3d views in the figures which are difficult to evaluate, while remotely operated vehicle (ROV) imagery, which the authors state provides ground-truthing, is not included in the figures.

Response: Thanks for highlighting the lack of these data. The bathymetry of the entire rift valley is now presented together with the sediment distribution in two separate figures in the supplementary (Figs. S4 and S5). Also, in Fig. 4 in the main text, several more detailed examples combining sediment thickness and bathymetry are presented from various locations within the rift valley. Three examples of detailed backscatter data are presented in supplementary figures S6, S7, and S8, together with accurate sediment thickness measurements and information from core locations. ROV imagery that provides ground truthing from various AVRs and other volcanic areas along the Mohns and Knipovich Ridges is being presented in a detailed volcanological study that we are currently writing up. We are planning to submit this paper, "Small and numerous volcanic eruptions documented from the ultraslow-spreading Arctic mid-ocean Ridges", within the next few months. Håvard Hallås Stubseid, Anders Bjerga, Ryan Portner, Haflidi Haflidason and Rolf Birger Pedersen are the authors. Therefore, ROV observations are not included here, and we have rephrased our sentences referring to these data.

Secondly, the chirp data tracklines were acquired at a spacing of 200-800 m (note it is not possible to evaluate the spacing since the tracklines are not plotted on the figures). In order to generate sediment isopach maps, the 2d sediment estimates were gridded at a resolution of 175 m (caption for Figure 1; Line 70). Since the grid spacing is potentially much less than the data spacing, the interpolation scheme (which is not specified, and needs more detail in any case) is likely to generate significant artifacts. The authors then use the interpolated sediment thickness grids as images to generate thickness statistics in Figure 3, which is likely to be problematic since the choice of interpolation scheme and grid resolution will impact the results. It would be better to analyze the population of individual sediment thickness estimates from the chirp profiles directly, without the interpolation.

Response: Thank you for assessing this potential problem. We have updated the information regarding this gridding and spotted an error in the reported line spacing. The spacing never exceeds 500m, and all survey lines are now illustrated in

supplementary Fig. S1. We gridded the sediment thickness to 175m to close the spacing between individual lines using the default weighted moving average without any further interpolation. The gridding was assessed by carefully evaluating backscatter data, core lengths, and accurate thickness measurements from chirp profiles. We find a near-perfect match between these data. See further description in Supplementary and supplementary Figs. S6, S7, and S8.

In terms of calculations, we have taken your advice and recalculated based on thicknesses directly from the chirp profiles. This has significantly improved our manuscript and supported our conclusions. Here, a total of ~1.1 million data points have been evaluated in Python. We also extracted slope data from 1m resolution bathymetry and removed all sediment thickness measurements associated with slopes above 30 degrees. Data analysis for the entire dataset is now updated and presented in Fig. 3. To investigate the differences between AVRs and areas outside, we have manually drawn polygons in QGIS and imported AVR boundaries into the sediment thickness dataset. We present the results from the data analysis of these two sub-areas in Fig. 3 in the main text. Details about the new approach for data analysis are presented in the supplementary information.

Thirdly, post-depositional processes, such as mass-wasting, current-driven erosion or redeposition, that likely significantly impact their sediment thickness, are not discussed. These processes have the potential to change the locus of sediment on the seafloor, especially in places like the Mohns ridge axial valley, where local basins and highs can lead to major differences in sediment accumulations. The authors need to address this uncertainty, possibly by examining local deep-water current patterns, and the potential effects that local depositional setting and seafloor slope could have in biasing their results (e.g. axial volcanic ridge flanks vs. local basins).

Response: A more thorough discussion regarding post-depositional processes, deep water currents, and topographic effects is now included in the supplementary. Both our seismic profiles and the collected gravity cores show a well-stratified sedimentary record deposited through steady hemipelagic sedimentation (see supplementary Figs. S2, S9, and S10). No gravity flows disturbing the sedimentation are observed. The well-preserved age-to-depth ratio in all dated cores indicates no disturbance from post-depositional processes. We also describe our observations related to the seafloor morphology along the Mohns Ridge, with no systematic relation between water depth, seafloor morphology, and sedimentation rate. We find examples of deep basins with and without thick sediments, as well as ridges and volcanic cones with varying degrees of sediment cover. This suggests that the measured sediment thickness results from hemipelagic sedimentation and not the local topography. However, we have excluded all thickness measurements associated with slopes above 30 degrees as these tend to not accumulate sediments. Unfortunately, no measurements of local deep-water current patterns are available from the Mohns Ridge. However, our calculation of sediment thicknesses and sedimentation rates from various locations in the rift valley in various topographic settings further strengthens our results.

Minor comments

L24 This width is thought to be more like 10 km by some (e.g. Katz et al., 2006), meaning that it's not very surprising to have volcanism anywhere in the axial trough.

Response: Agreed. We have updated and included the reference to Katz et al., 2006. Even though it might not be very surprising to have volcanism anywhere in the rift valley, no studies have yet documented the extent of volcanism outside AVRs and quantified the distribution of volcanism and the rejuvenation rate in the rift valley.

L25-28 This description is an over-simplification of what most think is happening at the axis on slow-spreading ridges and could benefit from some more references and details (e.g. Bickert et al., 2020; Reston, 2018)

Response: Agreed. We have updated to: **These ridges are characterized by intermittent volcanism and a lack of transform faults¹⁰ where a large part of the spreading is accommodated by tectonic activity rather than magmatic. The slowest spreading ridges are alternating between magmatic and amagmatic segments where large offset normal faults, or detachments, exhume lower crustal rocks and the mantle on the seafloor e.g.^{11, 12, 13.}**

L32 Missing reference to other detailed geochemical work, for example using using Po isotopes (Rubin et al., 1994)

Response: Agreed. We have added this and a paper using Ar/Ar at the Gakkel Ridge: O'Connor, J. M., Jokat, W., Michael, P. J., Schmidt-Aursch, M. C., Miggins, D. P., & Koppers, A. A. (2021). Thermochemical anomalies in the upper mantle control Gakkel Ridge accretion. *Nature communications*, 12(1), 6962.

L61, L109 what is "high resolution"?

Response: Specified to 1m resolution

L67 Needs additional caveats – assuming that no post-depositional reworking or mass wasting occurred

Response: See the answer to the comment above and a substantially edited discussion in supplementary regarding this.

L74 why compare to sedimentation rates in the equatorial MAR? Are there any other cores available on Mohns Ridge?

Response: Good suggestion. We have updated and rewritten the sentence and are now comparing the sedimentation rate within the rift valley with calculated rates in the open ocean of the Norwegian-Greenland Sea.

L77 How did you calculate the 30 cm vertical resolution of the chirp data? I would expect it to be much better than this since the instrument was on an AUV, bt impossible to check without better details in the supplement.

Response: The EdgeTech 2205 has a theoretical resolution of 6-10 cm. However, as 180 vertical meters were logged during the survey, the resolution of the upper meters of sediments is significantly affected. Therefore, by carefully assessing the chirp lines and measuring the thickness of individual reflectors, the true resolution is estimated to be 20-30 cm. The supplementary has been updated with all suggested details related to data collection and processing. We have also added a supplementary figure (Fig. S2) with selected chirp profiles where the reviewers can individually assess the true resolution of the data.

L83 define XRF

Response: Updated to X-ray fluorescence and added a reference in supplementary that further describes the applied XRF scanning method.

L105, L158 Here ROV ground truthing and visual observations are mentioned, but a detailed description of the images is missing, and images are not included

Response: See the answer to the comment above. As mentioned, ROV observations form part of a larger dataset for a more detailed volcanological study that we are currently writing up. These images and results are therefore not included here. Therefore, we have removed or rephrased sentences where ROV ground truthing or visual observations are mentioned.

L121-126 I can't easily verify that "that the volcanic activity is partly controlled by major normal faults" from the figures as presented

Response: Good point, thanks for addressing this. We have updated Fig. 4 with more detailed examples. Fig 4. d show a detailed picture of both the sediment distribution and the clean bathymetry where the reader can more easily see that a normal fault is clearly controlling the volcanic activity. On the eastern side of the fault, we see only very thin sediments, and therefore recent volcanic, whereas the western side is heavily sedimented without any volcanic eruptions for the last 50-100 Ka.

L185-189 the dating of basaltic dike intrusions is interesting, but would be better to be backed up with more than just an 'in prep' reference

Response: Agreed. This is part of a larger dataset where deep crustal rocks have been dated using a variety of geochronological methods. We have removed this paragraph from the current paper as this whole dataset will be presented in another manuscript that is currently under preparation.

L190-L198 This section about hydrothermal vents and mineral deposits is speculative and I'm not sure whether it's even relevant

Response: We agree that this section might to some extent be speculative. However, volcanic activity is a key factor for hydrothermal activity where magmatism is the main heat source for such activity. It is also recognized that volcanic eruptions can change the fluid pathways, disrupt the seafloor venting, or mask the deposits in lavas. Also,

hydrothermal activity and marine minerals are hot topics today, both in academia and the industry. Due to an increased demand for minerals and metals in our modern society, and to enable the green shift, parts of the world are now looking into deep-sea mining. This is currently an ongoing debate within the Norwegian government where there is a proposal to open for commercial activity within the Norwegian economic zone. The Norwegian Petroleum Directorate has, on behalf of the Ministry of Petroleum and Energy, released their resource estimate for marine minerals (unfortunately written in Norwegian; <https://www.npd.no/globalassets/1-mpd/fakta/havbunnsmineraler/publikasjoner/2023/ressursvurdering-havbunnsmineraler-20230127.pdf>)

With this in mind, we feel this section is highly relevant to include. This makes our findings even more applied and can be used also by governmental institutions and the industry. We hope that the reviewers and editors will agree that this section provides a significant contribution and is well-suited as part of the current manuscript. However, the section has been updated and further strengthened based on new calculations from the data analysis.

L201-203 This sentence is a bit of a stretch

Response: We feel that our paper provides a valuable contribution towards enhancing our understanding of crustal accretion and volcanic processes along ultraslow-spreading ridges. However, we have updated the sentence to: **The determination of volcanic seafloor ages from sediment thickness along the Mohns Ridge greatly expands our understanding of spreading dynamics and volcanic processes along the slowest spreading ridges of the global ridge system.**

This is the first study that quantifies the volcanism in different settings within a rift valley. Also, we provide valuable estimates for the volcanic rejuvenation of the rift valley floor. Moreover, we present the first age map of the upper lava flows in the rift valley providing valuable information regarding the spatial-temporal evolution of an ultraslow-spreading ridge. Therefore, we feel that our results greatly enhance our understanding of volcanism in ultraslow-spreading environments.

The Ms has a series of typos throughout, see examples on L85, L197 and L208 – there are likely more.

Response: These, and other typos have been corrected.

Availability of data and materials: I could not access the data via the link provided

Response: We are sorry that the provided link did not work. We updated the link and described the data availability in the main text as follows:

All acoustic data used in this study is available at the Norwegian Petroleum Directorate: <https://kartkatalog.geonorge.no/metadata/dyphavsdata/fe943f30-9a69-4c9c-9f35-726e28d9126f>

Because of extensive amounts of data, acoustic data is not currently available for direct download. However, the data can be ordered from the Norwegian Petroleum Directorate at a reasonable request as described here:

https://www.npd.no/globalassets/1-npd/om-oss/informasjonstjenester/karttjenester/bestilling-av-dyphavsdata_en.pdf

Figure 1

Location of chirp lines needs to be shown

Response: Thanks for the suggestion. We agree that the location of the chirp lines needs to be shown. To not disturb the dataset presented in Fig. 1, we have added the chirp lines in supplementary Fig. S1.

Need latitude and longitude marks on this and all other maps

Response: Thanks for pointing this out. Latitude and longitude are added to all large overview maps.

How are the AVRs picked? Are they really as linear/straight as shown?

Response: AVRs are picked based on their bathymetrical expression and are defined as bathymetric highs. We have updated the figure and outlined each AVR and added a number to each AVR based on their location along the Mohns Ridge.

How do the authors know that the faults (marked in red) are active? Are they that straight?

Response: Thanks for the input. Clearly, we do not know that these faults are active even though we suspect so. Figure legends have been updated and the term “active” has been removed to reflect this fact. The faults have been redrawn accurately based on bathymetry.

Inset location map is too small to be useful

Response: The inset location map has been slightly enlarged. Also, Norway and Iceland have been marked on the map.

Figure 3

Is this an analysis of the interpolated sediment thickness, or of the sediment thickness estimates themselves (i.e. without interpolation)? It should be the latter, since the interpolation is potentially misleading. What are the uncertainties?

Response: See the answer to the previous major comment where we explain that we now avoid using image analysis on the gridded surface and instead use your suggestion to do all calculations on the thickness exports directly from the chirp profiles.

Figures 4 & 5

I don't find these 3d perspective views very helpful in following the author's narrative

Response: Fig 4. has been updated with more detailed examples from our dataset to further highlight the results. This figure now includes a detailed picture of AVR-M13 showing the patchy age pattern we describe based on internal variations in the sediment distribution. We also show the same for a non-transform offset as well as a new inset for AVR-M14 where a normal fault clearly separated the volcanic activity.

However, we have chosen to keep Fig 5. in a 3D perspective. This is done to get a better look at the bathymetry and to highlight the young volcanic structures at the rift flank. From a volcanological perspective, we feel that this is a helpful figure that clearly illustrates one of our main findings.

References Cited

Bickert, M., Lavier, L., & Cannat, M. (2020). How do detachment faults form at ultraslow mid-ocean ridges in a thick axial lithosphere? *Earth and Planetary Science Letters*, 533, 116048. <https://doi.org/10.1016/j.epsl.2019.116048>

Katz, R. F., Spiegelman, M., & Holtzman, B. (2006). The dynamics of melt and shear localization in partially molten aggregates. *Nature*, 442(7103), 676–679. <https://doi.org/10.1038/nature05039>

Reston, T. J. (2018). Flipping detachments: The kinematics of ultraslow spreading ridges. *Earth and Planetary Science Letters*, 503, 144–157. <https://doi.org/10.1016/j.epsl.2018.09.032>

Rubin, K. H., Macdougall, J. D., & Perfit, M. R. (1994). 210po-210Pb dating of recent volcanic eruptions on the sea floor. *Nature*, 368(April), 841–844.

Response:

Reviewer #2 (Remarks to the Author):

The submitted manuscript “Volcanic evolution of an ultraslow spreading ridge” presents new data constraining the bathymetry, sediment thickness, and sedimentation rates along the Mohns Ridge. The authors use the data to demonstrate that the width of axial volcanism is greater than expected in this ultraslow setting. I found the manuscript to be well written and interesting, and consider the study an important contribution towards understanding the dynamics of crustal accretion at slow spreading ridges. I recommend its publication, with only minor revisions.

Response: We thank the reviewer for the detailed comments and positive evaluation of our manuscript.

My most significant comment is that I found it hard to independently assess some of the data interpretations, based on the current figures. In particular, I wonder if there is a way to make Figure 1 clearer, although I recognize the challenges of presenting different data types over a large area. I make more detailed suggestion on this point and others below.

Response: We have made this clearer by updating several figures and including more examples with clean bathymetry in Fig. 4. Also, we present additional data figures in the supplementary to support the interpretations. Regarding Fig. 1, see the detailed response to a comment below.

Detailed comments:

Lines 16–18: “We present the first age map of a mid-ocean ridge and find that nearly half of the 6-17 km wide inner rift valley floor has been rejuvenated by volcanic activity during the last 25 Kyr.”

I understand what the authors are trying to say here, but I do not think it is accurate to say this is the first age map of a mid-ocean ridge. Magnetic anomaly maps could be considered age maps, and there have also been other published studies with multiple dated samples along a ridge segment. For example:

Connor et al., Nature Communications, 2021, Thermochemical anomalies in the upper mantle control Gakkel Ridge accretion.

Baines et al., EPSL, 2008, The rate of oceanic detachment faulting at Atlantis Bank, SW Indian Ridge.

Schwartz et al., Science, 2005, Dating the growth of oceanic crust at a slow-spreading ridge.

Response: That is a very good point, and we see that this statement might be misunderstood. We have rewritten now stating that this is an age map of the upper lava flows within the rift valley.

Line 23: There is a typo, replace “were” with “where”.

Response: Done

Line 42: It would be helpful to describe the location of the ridge, for non-specialist readers.

Response: Agreed. Updated to: In this contribution, we focus on the 500 km long oblique spreading Mohns Ridge, located in the Norwegian-Greenland Sea, between 71.2°N and 73.5°N (Fig. 1).

Lines 65–68: Given that this text refers to the age data, the authors could consider moving it below the discussion of their age data (lines 71–78).

Response: Agreed. This paragraph has been moved to the end of the section where we discuss the ages.

Lines 106–108: “Individual volcanic centers can be seen in the isopach map as small, isolated fields with varying sediment cover leaving a patchy age pattern (Fig. 4).”

These are not apparent to me in Figure 4. From the subsequent text, it sounds like this observation is based on a detailed study that the authors are currently writing up, but this does not seem to me to be supported by the data in this manuscript.

Response: Agreed. Fig. 4 has been updated to highlight these details and enable the reader to assess this statement independently. This is now hopefully possible to see both in Fig. 4a and b. Especially in b, where we have highlighted two volcanic centres with varying degrees of sediment cover. Details from our other study are removed and will, as you assume, be part of another manuscript we are currently writing up.

Line 122: There is a typo, replace “nucleates” with “nucleate”.

Response: Done

Line 142: There is a typo, replace “occurs” with “occur”.

Response: Done

Lines 145–146: “The isopach map reveals that for the volcanic seafloor that formed during the last 25 Kyr, as much as 50% formed outside the AVRs.”

This is hard to assess based on the current figures (please see my comments below).

Response: Thanks for addressing this. We completely agree, and Fig. 1. has been updated and the outline of each AVR has been marked as a red line. This makes it easier for the reader to get the impression that a large part of the volcanic activity occurs outside these bathymetric highs.

Figure 1: The manuscript figures are very high quality, but I wonder if there is a clearer way to present the combined bathymetric and sediment thickness data. The AVRs show up very nicely on the bathymetry, as seen in Figure 2. It would be useful to be able to clearly identify those and other bathymetric features in Figure 1, but it is difficult because the sediment thickness data masks the bathymetry. I suspect the authors have already experimented with this, but would it be clearer to show the bathymetry as a grayscale hillshade, then overlay that with a semitransparent color scale for sediment thickness? This might also more clearly accentuate variations in the sediment thickness, since it is difficult to see subtle differences on the current grayscale. It would also be helpful to number the AVRs in this figure, so the reader can more easily relate the descriptions in the text to the figure. Finally, I would recommend adding a label for Norway and Iceland to the location map, to provide context.

Response: This figure has been one of our main challenges during the study. We have experimented with numerous variations to highlight our findings. We have tried different color schemes for both bathymetry and sediment distribution. Also, we have tried variations in transparency and different ways to drape the sediments on top of bathymetry. However, we concluded that the way it is currently presented gives the most detailed and correct image even though it to some degree compromises the underlying bathymetry. Other variations of the figure where the bathymetry is less masked make it even more difficult to see the variations in the sediment thickness. As the sediment is the most important thing to visualize, we chose to keep the figure in its current form. However, we have made some adjustments according to your suggestions. The outline of each AVR has been marked with a red line to highlight these bathymetric highs. All AVRs have also been numbered according to their locations along the ridge, and we have added a label for Norway and Iceland in the inset map.

We have solved this issue with sediments vs. bathymetry on the figure, by creating two supplementary figures (Figs. S4 and S5) where clean bathymetry is presented alongside the sediments for both areas.

Figure 2. I would change the “processed sediment cores” label in the key to “undated cores”. My initial interpretation (before reading the figure caption) was that these were cores that the authors processed for geochron., but which did not yield data, which is incorrect.

Response: Thanks for the suggestion and we see the misunderstanding. These “processed sediment cores” are cores that have been photo scanned and chemically analyzed without any direct C¹⁴ dating. However, their stratigraphy has been correlated with other cores with and without C¹⁴ dating providing an indirect dating. Therefore, we update the term to “stratigraphically correlated sediment cores”.

Overall, this is an interesting study and a well written manuscript, with high quality figures.

I hope these comments are useful.

Response: Thank you so much for your overall positive feedback on our manuscript. Comments from both reviewers have been very useful and helped significantly improve our manuscript.

REVIEWERS' COMMENTS

Reviewer #1 (Remarks to the Author):

In this revised manuscript, Stubseid et al. present a detailed study of volcanism and sedimentation on the ultraslow-spreading segment of the Mohns Ridge, using a combination of near-bottom geophysical data and carbon isotope dating of foraminifera in sediment cores. I am glad to see that the authors have made extensive revisions to the text and figures, which result in a greatly improved manuscript. The figures are now much improved, and I am very glad to see the tracklines in Figure 1. The additional information provided about data acquisition in the Supplementary material is very useful, and I'm glad to see the chirp images themselves being provided in Figure S2. I think the Ms should be accepted after the authors deal with the minor issue I describe below.

My only concern remains due to the overlooked effects of mass wasting and bottom current activity, which were not addressed in the revised Ms. For example, the authors find deep basin lows with both thick and thin/absent sediment, which makes one wonder whether those without sediment have been winnowed or eroded completely. In order to partly overcome this problem, it would be useful to understand the prevailing modern current patterns, to check whether there is a bias in sediment accumulation on the leeward side of seafloor slopes, for example. Possible ways to accomplish this would be to look at shipboard ADCP data from the region, or use hindcast data extracted from a global model such as HYCOM (Bleck, 2002). A similar issue was recently considered on the East Pacific Rise, where constraints the relative age of volcanism was inferred using sediment thickness as a proxy for volcanic exposure age, in the presence of weak bottom current activity (Fabrizzi et al., 2022). If the currents are found to be weak in modern day models and data, it would strengthen the authors case, but needs to be mentioned in the main text discussion. Either way, these caveats and considerations need to be included in both the introduction and in the discussion.

Minor comments

L8 delete 'the' before 'ocean crust'

L14 I think Carbon-14 or ^{14}C is more widely used

L15 change 'and reveal' to 'to reveal'

L16 This isn't the 'first' age map of upper lava flows...perhaps change to 'an age map...'

L22 change 'drift apart' to 'spread apart'

L30 change 'magmatic' to 'magmatism'

L31 change 'are alternating' to 'alternate'

L34 awkward wording 'during the last decades'

L35 choose a style for units of rates, either mm yr⁻¹ or cm/ka

L84 need to add caveats here about mass-wasting, sediment winnowing by bottom current activity, any terrestrial input, and possible carbonate dissolution

L97 'This' needs a subject

L129 just put the %, not 'more than 60%'

L134 The use of the term 'half life' here is potentially confusing. I know what you mean, and understand if you want to retain it, but I think it might confuse readers.

L161 change constructions to constructs

L202 Need discussion of the influence of bottom currents, mass wasting here

L259 This 'what' suggests...

L264 what does the term 'eruption cell' mean?

References Cited

Bleck, R. (2002). An oceanic general circulation model framed in hybrid isopycnic-Cartesian coordinates. *Ocean Modelling*, 4(1), 55–88. [https://doi.org/10.1016/S1463-5003\(01\)00012-9](https://doi.org/10.1016/S1463-5003(01)00012-9)

Fabrizzi, A., Parnell-Turner, R., Gregg, P. M., Fornari, D. J., Perfit, M. R., Wanless, V. D., & Anderson, M. (2022). Relative Timing of Off-axis Volcanism from Sediment Thickness Estimates on the 8°20'N Seamount Chain, East Pacific Rise. *Geochemistry, Geophysics, Geosystems*, 1–21. <https://doi.org/10.1029/2022gc010335>

Reviewer #1 (Remarks to the Author):

In this revised manuscript, Stubseid et al. present a detailed study of volcanism and sedimentation on the ultraslow-spreading segment of the Mohns Ridge, using a combination of near-bottom geophysical data and carbon isotope dating of foraminifera in sediment cores. I am glad to see that the authors have made extensive revisions to the text and figures, which result in a greatly improved manuscript. The figures are now much improved, and I am very glad to see the tracklines in Figure 1. The additional information provided about data acquisition in the Supplementary material is very useful, and I'm glad to see the chirp images themselves being provided in Figure S2. I think the Ms should be accepted after the authors deal with the minor issue I describe below.

Response: Thank you for your positive feedback. We are glad that you appreciate our substantial revision of both the text and figures and thank you again for your input to improve our paper.

My only concern remains due to the overlooked effects of mass wasting and bottom current activity, which were not addressed in the revised Ms. For example, the authors find deep basin lows with both thick and thin/absent sediment, which makes one wonder whether those without sediment have been winnowed or eroded completely. In order to partly overcome this problem, it would be useful to understand the prevailing modern current patterns, to check whether there is a bias in sediment accumulation on the leeward side of seafloor slopes, for example. Possible ways to accomplish this would be to look at shipboard ADCP data from the region, or use hindcast data extracted from a global model such as HYCOM (Bleck, 2002). A similar issue was recently considered on the East Pacific Rise, where constraints the relative age of volcanism was inferred using sediment thickness as a proxy for volcanic exposure age, in the presence of weak bottom current activity (Fabbrizzi et al., 2022). If the currents are found to be weak in modern day models and data, it would strengthen the authors case, but needs to be mentioned in the main text discussion. Either way, these caveats and considerations need to be included in both the introduction and in the discussion.

Response: We thank you for addressing this concern again. This has been discussed in the Supplementary Information, in which we use both chirp profile data and sediment cores to support the concept of constant hemipelagic sedimentation. However, we have now also induced such a discussion in the main text when we first present our age model. Unfortunately, no bottom current measurements exist, and we do not have any shipboard ADCP data to characterize the currents.

The paragraph in the main text is updated to: After sediments are deposited, they can be reworked through mass wasting or by bottom current activity. To avoid bias due to the effects of gravitational mass wasting we exclude all sediment thickness measurements associated with slopes above 30 degrees. There are no signs of mass wasting/slumping within our chirp profiles (Fig. S2) or gravity cores (Figs. S9 & S10) suggesting that the sediment cover is a result of steady hemipelagic sedimentation. To our knowledge, there are no measurements of bottom currents along the Mohns Ridge. Only intermediate currents in the water column have been studied e.g. ³⁵ but their effect on the seafloor remains poorly understood. Also, modern-day current patterns may not be

representative from a geological time perspective. The chirp profiles document that sediments are evenly draped on top of the seafloor topography (Fig. S2) with no systematic differences in the degree of sedimentation between local basins and bathymetric highs indicating weak bottom currents. As we show that the hemipelagic sediment accumulates at a relatively constant rate, at least for the last 50 Kyr, (Fig. 2 and Supplementary Information), the sediment thickness reflects the age of the underlying volcanic flows³⁶. The isopach map (Fig. 1), accordingly, reflects the spatial-temporal evolution of volcanic activity within the rift valley.

Minor comments

L8 delete 'the' before 'ocean crust'

Response: Done

L14 I think Carbon-14 or ¹⁴C is more widely used

Response: Agreed. All updated to ¹⁴C

L15 change 'and reveal' to 'to reveal'

Response: Done

L16 This isn't the 'first' age map of upper lava flows...perhaps change to 'an age map...'

Response: As this approach has not previously been applied to large-scale age mapping of the rift valley floor along an entire ridge segment, we have updated to: Here, we present the first age map of the upper lava flows within the rift valley of a mid-ocean ridge and find that nearly half of the rift valley floor has been rejuvenated by volcanic activity during the last 25 Kyr.

L22 change 'drift apart' to 'spread apart'

Response: Done

L30 change 'magmatic' to 'magmatism'

Response: Done

L31 change 'are alternating' to 'alternate'

Response: Done

L34 awkward wording 'during the last decades'

Response: Updated to: for several decades

L35 choose a style for units of rates, either mm yr⁻¹ or cm/ka

Response: We have used mm yr⁻¹ for spreading rates, as it is a standard way to present this. For sedimentation rates, we use cm/Ka.

L84 need to add caveats here about mass-wasting, sediment winnowing by bottom current activity, any terrestrial input, and possible carbonate dissolution

Response: Done. We have added a section where we use our data and observations to discuss the potential influence of mass wasting and bottom currents.

L97 ‘This’ needs a subject

Response: Updated to: **these thickness measurements imply....**

L129 just put the %, not ‘more than 60%’

Response: Done

L134 The use of the term ‘half life’ here is potentially confusing. I know what you mean, and understand if you want to retain it, but I think it might confuse readers.

Response: We see your point but choose to stick to the formulation as we find this term effective in describing our observed dynamic age pattern at AVRs.

L161 change constructions to constructs

Response: Done

L202 Need discussion of the influence of bottom currents, mass wasting here

Response: This has been solved in the result chapter where we first present our age model. Here, we include relevant observations regarding the potential effects of mass wasting and bottom currents and argue that sediments are evenly draped on the seafloor topography as a result of steady hemipelagic sedimentation.

L259 This ‘what’ suggests...

Response: Updated and aggregated with the previous sentence

L264 what does the term ‘eruption cell’ mean?

Response: Updated to “eruptive areas**”. These are the small and local places where we observe “recent” volcanism younger than surrounding areas. From the current dataset, we cannot conclude whether these areas represent one or several individual eruptions. However, based on additional data that is currently being written up as part of a different paper, we have indications that these eruptive areas consist of multiple small eruptions.**

References Cited

Bleck, R. (2002). An oceanic general circulation model framed in hybrid isopycnic-Cartesian coordinates. *Ocean Modelling*, 4(1), 55–88. [https://doi.org/10.1016/S1463-5003\(01\)00012-9](https://doi.org/10.1016/S1463-5003(01)00012-9)

Fabbrizzi, A., Parnell-Turner, R., Gregg, P. M., Fornari, D. J., Perfit, M. R., Wanless, V. D., & Anderson, M. (2022). Relative Timing of Off-axis Volcanism from Sediment Thickness Estimates on the 8°20’N Seamount Chain, East Pacific Rise. *Geochemistry, Geophysics, Geosystems*, 1–21. <https://doi.org/10.1029/2022gc010335>